# Vitamin D and Calcium in Osteoporosis, and the Role of Bone Turnover Markers: A Narrative Review of Recent Data from RCTs

**DOI:** 10.3390/diseases11010029

**Published:** 2023-02-08

**Authors:** Gavriela Voulgaridou, Sousana K. Papadopoulou, Paraskevi Detopoulou, Despoina Tsoumana, Constantinos Giaginis, Foivi S. Kondyli, Evgenia Lymperaki, Agathi Pritsa

**Affiliations:** 1Department of Nutritional Sciences and Dietetics, School of Health Sciences, International Hellenic University, 57400 Thessaloniki, Greece; 2Department of Clinical Nutrition, General Hospital Korgialenio Benakio, 11526 Athens, Greece; 3Department of Nutritional Science and Dietetics, University of the Peloponnese, 24100 Kalamata, Greece; 4Department of Food Science and Nutrition, School of Environment, University of the Aegean, 81400 Myrina, Greece; 5Department of Biomedical Sciences, International Hellenic University, 57400 Thessaloniki, Greece

**Keywords:** osteoporosis, vitamin D, calcium, bone density, bone turnover markers, PTH, falls, fractures

## Abstract

Osteoporosis is a common disease, defined primarily by a low measured bone density, which is associated with an increased risk of fragility fractures. Low calcium intake and vitamin D deficiency seem to be positively correlated with the prevalence of osteoporosis. Although they are not suitable for the diagnosis of osteoporosis, the biochemical markers of bone turnover can be measured in serum and/or urine, enabling the assessment of the dynamic bone activity and the short-term effectiveness of the osteoporosis treatment. Calcium and vitamin D are essential for maintaining bone health. The aim of this narrative review is to summarize the effects of vitamin D and calcium supplementation separately and in combination, on bone density and circulating serum and blood plasma vitamin D, calcium, parathyroid hormone levels, markers of bone metabolism concentrations, and clinical outcomes, such as falls and osteoporotic fractures. We searched the PubMed online database to find clinical trials from the last five years (2016–April 2022). A total of 26 randomized clinical trials (RCTs) were included in this review. The present reviewed evidence suggests that vitamin D alone or in combination with calcium increases circulating 25(OH)D. Calcium with concomitant vitamin D supplementation, but not vitamin D alone, leads to an increase in BMD. In addition, most studies did not detect significant changes in circulating levels of plasma bone metabolism markers, nor in the incidence of falls. Instead, there was a decrease in blood serum PTH levels in the groups receiving vitamin D and/or Ca supplementation. The plasma vitamin D levels at the beginning of the intervention, and the dosing regimen followed, may play a role in the observed parameters. However, further study is needed to determine an appropriate dosing regimen for the treatment of osteoporosis and the role of bone metabolism markers.

## 1. Introduction

Osteoporosis is a common condition, defined primarily on the basis of bone density assessment, and it is associated with an increased risk of fragility fractures. There are two types of osteoporosis, depending on the etiology of its development: primary and secondary. Primary osteoporosis is defined as osteoporosis that occurs after menopause (postmenopausal osteoporosis), or with increasing age (senile osteoporosis). Secondary osteoporosis is caused by various disorders and drug side effects [1]. Secondary causes of bone loss, which include adverse effects of drug therapy, endocrine disorders, eating disorders, kidney disease, and cancer, among others, often go undiagnosed in patients with osteoporosis [2].

The clinical significance of osteoporosis is related to a decrease in bone strength, leading to an increased risk of fragility fractures from low energy trauma [1]. An osteoporotic fracture (OF) occurs every three seconds, and 8.9 million osteoporotic fractures are reported to occur annually worldwide [3]. The estimated number of people >50 years of age suffering from osteoporosis in Europe in 2019 was around 32 million, of which 25.5 million were women and only 6.5 million were men [4]. In addition, the estimated number of people who suffered an osteoporotic fracture in 2019 was 4.28 million, while in 2034, this number is estimated to reach 5.34 million [4]. The highest prevalence of osteoporosis was reported in Africa, with 39.5% (95% confidence interval, CI 22.3–59.7) [5].

### 1.1. Risk Factors of Osteoporosis

The risk of OF results from a combination of factors that act mainly through the reduction in bone mineral density (BMD), as well as from factors that are completely or partially independent of it [1]. Clearly, this distinction is not clear-cut, and several risk factors act simultaneously through different mechanisms [1]. Many risk factors are associated with OF, such as low maximum bone mass, hormonal factors, the use of certain medications (e.g., glucocorticoids), smoking, low physical activity, a low dietary intake of calcium (Ca) and vitamin D, race, small body size, and an individual or family history of fracture [6]. In individuals where multiple risk factors coexist, the risk of fracture is higher than in individuals with a single risk factor [1]. In addition, the presence of comorbidities increases the risk of fracture. Therefore, all risk factors and co-morbidities should be taken into account when assessing the risk of fracture, and when determining whether further treatment is needed [6].

Vitamin D and calcium are essential for maintaining bone health, and their deficiency is an important risk factor for the development of osteoporosis [7]. Vitamin D is naturally produced in human skin upon exposure to sunlight. Due to a modern lifestyle, there is insufficient exposure to sunlight for the synthesis of vitamin D [8], resulting in particularly low seasonal concentrations of 25-Hydroxy-Vitamin D3 (25(OH)D3), usually during winter and spring [9]. The determination of normal and insufficient levels for serum vitamin D constitutes a highly debated topic [10,11]. While there is an agreement that values of 25(OH)D < 10 ng/mL (=25 nmol/l) represent a state of severe deficiency, there is no consensus on what would be considered as a “normal” value [10,11]. The Italian Osteoporosis Society suggests that levels >30 ng/mL are considered adequate, while levels of between 30 and 50 ng/mL (=75 and 125 nmol/l) are considered optimal [1].

Calcium is essential for many body functions, such as building and maintaining healthy bones, while vitamin D can help with the intestinal absorption of ingested calcium [12]. Calcium is a key structural component of bone minerals, which ensures skeletal health. It is ingested though diet or supplements, and is excreted in the urine [13]. If the daily calcium intake is reduced, a negative calcium balance occurs with secondary hyperparathyroidism, with detrimental consequences for the body [1].

### 1.2. Diagnosis of Osteoporosis

The measurement of BMD is sufficient for the diagnosis of osteoporosis (diagnostic threshold), but the combined assessment of BMD with independent risk factors is necessary to identify individuals that are at a high risk of fracture, requiring specific pharmacological or non-pharmacological intervention, depending on the type of risk factors that are likely to be involved [1,14]. Bone strength depends on both quantitative factors, such as BMD, assessed via dual-energy X-ray absorptiometry (DXA), and qualitative factors, such as geometry, microstructure, and the mineral and organic composition of the matrix, which are not usually evaluated in clinical practice [1].

DXA is the most widely used bone densitometry technique that is considered the gold standard reference [14]. According to the World Health Organization (WHO) criteria, osteoporosis is defined as BMD that is 2.5 standard deviations (SD) or more below the mean value for young, healthy women (T-score < −2.5 SD), while severe osteoporosis is defined as a T-score below −2.5 SD with one or more fragility fractures [15].

In addition, computed tomography (CT) and peripheral quantitative computed tomography (pQCT) are 3D techniques that use X-ray attenuation to construct images of bone [16]. For the measurement of the spine and other areas, a full-size CT scanner is mainly used, while special pQCT scanners are used to measure the radius, tibia, or the anus of the femur [16].

### 1.3. Biomarkers

Although biochemical markers of bone turnover cannot be used to diagnose osteoporosis, they can be measured in serum and/or urine, allowing a dynamic assessment of bone activity and health [17,18]. Moreover, they have been shown to reflect the effectiveness of osteoporosis treatment in a relatively short period of time [19]. In addition, alterations in bone turnover indices may indicate accelerated bone loss or other primary or secondary bone disorders [1]. The available biochemical markers are classified into bone formation markers, which include bone alkaline phosphatase (bALP), osteocalcin (OC), procollagen type I aminotellarpropeptide (P1NP), and bone resorption markers, which include deoxypyridoline (DPD), pyridinoline, N- and C-terminal telopeptides of type I collagen (NTx and CTx, respectively), and which have some prognostic significance for fracture [1,14]. As has already been underlined by the European Food Safety Authority (EFSA) in 2016, more research is needed to delineate the relationship between bone turnover markers and vitamin D status [20]. Moreover, due to the large analytical variation in measuring methods and biological variability, which affects their use in individual patients, they cannot be used for routine clinical evaluations yet [1].

Calcium phosphate homeostasis is mainly regulated by vitamin D, parathyroid hormone (PTH) and fibroblast growth factor 23 (FGF23) [21]. Vitamin D deficiency results in a reduced absorption of calcium and phosphate from the gut, which then increases the absorption of these ions by the bones, increasing PTH production [22]. An inverse association of circulating PTH levels with calcium intake is generally observed, implying that PTH decreases with increasing dietary calcium intake [13].

### 1.4. Treatment

Several effective treatments, along with pharmacological interventions, can lead to a reduction in fracture risk in postmenopausal women [14]. Diet and physical activity are the primary modifiable factors associated with bone health, although other lifestyle and environmental factors may also play an important role [16]. Optimal vitamin D status promotes skeletal health and is recommended as a specific treatment in people at high risk for fragility fractures [23]. Adequate amounts of calcium are also required for the mineralization of fractures, and the reduced bone recovery often observed in osteoporotic patients can be attributed to calcium and vitamin D deficiencies. Therefore, calcium and vitamin D supplementation is widely recommended for the prevention of osteoporosis and subsequent fractures in osteoporotic patients [7]. More specifically, recommendations for postmenopausal women for osteoporosis management include a daily calcium intake of 800–1200 mg from diet and/or supplements (if dietary intake is insufficient), along with dietary protein ingestion [14]. A daily dose of 800 IU vitamin D is recommended for postmenopausal women who are at an increased risk of fracture, and those having vitamin D insufficiency [14].

However, the recent data are quite contradictory. Some studies showed that calcium supplementation combined with vitamin D reduced the risk of fractures [24,25,26], while others showed no effect [21,27,28,29,30]. A meta-analysis suggested that concomitant supplementation with calcium and vitamin D led to a significantly reduced risk of total fractures by 15% (summary relative risk estimates, SRRE: 0.85; 95% CI, 0.73–0.98) and a 30% reduced risk of hip fractures (SRRE: 0.70; 95% CI, 0.56–0.87) [31]. In addition, the dosage regimen seems to play a role. A meta-analysis concluded that the combination of calcium and vitamin D significantly increased femoral neck BMD only when the dose of vitamin D intake was less than 400 IU/day (standard mean differences, SMD = 0.335; 95% CI: 0.113 to 0.558), compared to higher doses (SMD = −0.098; 95% CI: −0.109 to 0.305), whereas calcium had no effect on femoral neck BMD [12]. In addition, it was observed that only dairy products fortified with calcium and vitamin D, and not the combination of calcium and vitamin D supplements, had a significant effect on total BMD (SMD = 0.784; 95% CI: 0.322 to 1.247) and lumbar spine BMD (SMD = 0.320; 95% CI: 0.146 to 0.494) [12].

The aim of this review was to summarize recent data from the supplementation of vitamin D and calcium as sole compounds or a combination, regarding their effects on bone density, circulating serum and blood plasma levels of vitamin D, calcium, parathyroid hormone, and bone metabolism markers, as well as clinical outcomes, such as falls and OF.

## 2. Methods

The PubMed online database was searched. Three different search strategies were used to find clinical trials from the last five years (2016-April 2022) in three different directions. A search was performed on keywords that were joined by Boolean operators in the MeSH vocabulary and in Titles/Summaries. More specifically, the search strategies “(osteoporosis[MeSH Terms] OR osteoporosis[Title/Abstract] OR bone mineral density[Title/Abstract]) AND (vitamin D[MeSH Terms] OR cholecalciferol[MeSH Terms] OR ergocalciferol[MeSH Terms]) AND (clinical trial* OR randomised trial) NOT (review)” were used for vitamin D and osteoporosis, “(osteoporosis[MeSH Terms] OR osteoporosis[Title/Abstract] OR bone mineral density[Title/Abstract]) AND (calcium[MeSH Terms]) AND (clinical trial* OR randomised trial) NOT (review)” for calcium and osteoporosis, and “(osteoporosis[MeSH Terms] OR osteoporosis[Title/Abstract] OR bone mineral density[Title/Abstract]) AND (calcium[MeSHTerms]) AND (clinical trial* OR randomised trial) AND (vitamin D[MeSH Terms] OR cholecalciferol[MeSH Terms] OR ergocalciferol[MeSH Terms]) NOT (review)” for the combination of vitamin D and calcium in osteoporosis.

The inclusion criteria for studies were (1) that they would be randomized clinical trials (RCTs); (2) that the interventions involved only vitamin D supplementation, calcium supplementation, or a combination of the above; and (3) that they were conducted in an adult population. Studies with only per os supplementation in supplement form were included. More specifically, studies with supplemented/fortified foods were excluded from the present analysis, since foods contain other nutrients that may affect bone density. Studies in people with disorders that may affect bone and calcium metabolism (e.g., chronic kidney disease, HIV, pregnancy, thyroid, glucocorticoid use, and the use of antiepileptic and antidepressant drugs) and those that were not in English were also excluded. In addition, studies in which the intervention and comparison groups did not allow for the assessment of the independent contribution of vitamin D or calcium to outcome were excluded. For example, when these supplements were taken in a multivitamin or were used as part of a multimodal intervention that included other pharmacological agents, or environmental or behavioral interventions.

A search on vitamin D and osteoporosis resulted in 259 studies, 17 of which met the inclusion criteria. A search strategy on calcium and osteoporosis identified 114 studies, but none of them met the inclusion criteria. Finally, from the search strategy on vitamin D, calcium, and osteoporosis, 67 studies were detected, from which 9 were included in this review as they met the inclusion criteria. A total of 26 studies were finally included in this review.

## 3. Results

### 3.1. Vitamin D Supplementation

In total, 17 studies involving intervention with vitamin D supplements were identified, which are presented in Table 1. Participants’ (n = 4482) mean age varied from 43 to 75 years, and the durations of the studies ranged from 8 weeks to 3 years [18,32,33,34,35,36,37,38,39,40,41,42,43,44,45,46]. In seven studies, only females were included [18,32,33,34,35,36,37], in two studies, only males were included [38,39], while in the remaining eight studies, both sexes were included [40,41,42,43,44,45,46]. Studies were conducted in the USA [32,35,36], Canada [41,42], Brazil [18,34], European countries [33,37,38,40,46], New Zealand [45], and Asia [43,44]. Participants’ health statuses differed from study to study. In some clinical trials, participants were healthy individuals [38,43,47]; while in others, postmenopausal [18,33,35,36] or elderly women [37], and subjects with vitamin D deficiency [32,44,46] or/and low BMD [34]. Furthermore, some studies included patients with osteoporosis [40], and patients in an intensive care unit (ICU) [46] or other categories [41,42]. The dose of vitamin D ranged from 400 IU daily to 10,000 IU daily [41,42].

#### 3.1.1. Vitamin D and BMD

Vitamin D supplementation has been generally considered beneficial for the prevention and treatment of osteoporosis. However, the results are controversial. Of the 17 clinical studies, most found no difference [38,40,46], two found a decrease in BMD [13,37], and only two documented an increase in bone density or related variables, such as the bone mineral balance between the supplementation and control group [39,43]. More specifically, in the study of Rangarajan et al., 16 young males (18–35 years) with vitamin D deficiency participated. They were treated with 60,000 IU of vitamin D weekly for 3 years, and the bone mineral changes were assessed with the use of isotopes [39]. The study concluded that the mean increase in the bone mineral balance of subjects receiving the intervention was 0.04 ± 0.05, compared to a net negative value for the control group (−0.03 ± 0.01) [39]. Within the supplemented group, the subgroup with low dietary Ca intake showed a higher Ca isotope enrichment, resulting in a significant change in BMD compared to the control group [39]. A randomized clinical trial (48.5% women) of 8 week supplementation with vitamin D (50,000 IU weekly) in healthy adults of 20–60 years old showed that the changes in T-score in the intervention group were significantly greater than those that were observed in the control group (0.81 vs. 0.30, respectively, *p* < 0.001) [43]. In addition, the Z-score changes in the intervention group were significantly greater than in the control group [43].

#### 3.1.2. Vitamin D and Serum 25(OH)D

Fourteen studies measured serum 25(OH)D levels in blood before and after the intervention. All studies reported a significant increase in serum blood levels [13,18,33,34,36,37,38,39,40,41,42,44,46,48] except in one study, which was designed to achieve a standard concentration of vitamin D [35]. It is noted that when seasonality was taken into account, higher plasma 25(OH)D levels were observed during July–September, compared to January–March (*p* < 0.001) [38].

The dosing regimen of vitamin D appears to influence the overall outcome and markers of osteoporosis. In the study of Burt et al., mean serum 25(OH)D measurements did not change for the 400 IU group from baseline to 36 months (76.3–77.4 nmol/L), whereas a significant increase was observed in the 4000 IU group in the first 3 months (81.3–115.3 nmol/L), with a further increase to 132.2 nmol/L by month 36 [41]. An increase was also observed from baseline to 3 and 18 months in the 10000 IU group (78.4–188.0–200.4 nmol/L), but then, 25(OH)D was decreased to 144.4 nmol/L at 36 months [41]. However, such a high dose of vitamin D administration was associated with significantly greater bone loss, possibly due to a reduction in plasma PTH [41].

#### 3.1.3. Vitamin D and PTH

Of the eight studies that also measured plasma PTH levels [18,33,34,35,38,39,40,41], one detected no significant change in PTH between the supplementation and placebo group [34], while all others observed a decrease in PTH concentration. Indeed, an RCT in 160 postmenopausal women identified a significant decrease in PTH levels by 21.3% in the supplementation group, before and after the intervention (*p* < 0.001), while a small increase of 8.5% was found in the placebo group before and after the intervention (*p* = 0.493), with a significant difference between the two groups (*p* = 0.002) at the end of the study [18]. Similarly, an RCT in men concluded that plasma PTH concentration changed from 64.2 ± 23.9 at baseline, to 43.1 ± 13.7 pg/mL in the intervention group [39]. In addition, based on seasonality, higher levels were observed in the January–March period, compared to the April–June period (*p* = 0.006) [38].

#### 3.1.4. Vitamin D and Falls

Regarding the incidents of falls, five clinical trials were detected, which included this parameter in the assessment outcomes of vitamin D intervention. All [37,40,41,48] but one [34] reported no significant changes in falls. One study reported a higher incidence of falls in the intervention group [34]. More particularly, in that study, the adjusted risk ratio of the rate of falls and recurrent falls was 1.95-fold (95%CI, 1.23–3.08) and 2.8-fold (95%CI, 1.43–5.50) higher, respectively, in women in the placebo group, compared with the vitamin D supplementation group (*p* = 0.003) [34].

#### 3.1.5. Vitamin D and Bone Turnover Markers

A total of eight studies measured the concentration of bone turnover markers in the plasma or serum of participants [18,35,38,41,42,44,46,48,49]. Of these, seven found no significant effect of vitamin D on plasma bone turnover markers between the intervention and control groups [18,35,42,44,46,48,49], while the study by Burt et al. showed an increase in CTx at all time points within the groups; however, without reporting whether these differences were significant [41]. In this study, women in the 10,000 IU group had higher CTx values than the 400 IU group, without this difference being significant (*p* = 0.507) [41], while in another vitamin D supplementation study, no significant effect on bALP, CTX, OC, and P1NP values was observed [38]. However, analyzing gender differences, OC in men was 11.8 ng/mL (95%CI 8.8–13.9) at baseline and 12.1 ng/mL (95%CI 9.1–15.6) at follow-up, resulting in a mean treatment effect of 0.131 (95%CI 0.020–0.242, *p* = 0.022) [49]. Similarly, in ICU, patients’ CTX decreased both over time within groups (*p* < 0.001) and OC increased both over time (*p* < 0.001) [46], while between-group differences were not significant [46]. Nahas-Neto et al. (2018) observed a significant decrease between time points in serum carboxy-terminal collagen crosslinks (s-CTX) (−24.2%) (*p* < 0.0001) and P1NP (−13.4%) (*p* < 0.003) only in the group receiving vitamin D supplementation, while no changes were observed between the time points in the placebo group (s-CTX, −6.9%, *p* = 0.092 and P1NP (−0.6%, *p* = 0.918) and between the intervention and placebo groups (s-CTX *p* = 0. 913 and P1NP *p* = 0.254) [18]. A 20-week intervention found that cholecalciferol supplementation increased the bALP concentration (+1.7 ± 1.9 μg/L) significantly more than the placebo (+1.1 ± 1.7 μg/L, *p* = 0.004), but this did not have the same effect for the PINP, β-isomerized C-terminal telopeptides (β-CTx), or tartrate-resistant acid phosphatase (TRAP5b) [44].

**Table 1 diseases-11-00029-t001:** Characteristics of included randomized controlled trials with Vitamin D supplementation.

Authors	Country	Study Design	n	Sex (W%)	Age (Mean)	Sample (at Baseline)	Groups	Duration of Intervention	Follow-Up	Dose of Vitamin D, Frequency	Effects on BMD	Effects on 25(OH)D and PTH	Effects on Bone Turnover Indices	Effects on Falls	Other Results
[18]	Brazil	Double-blind, placebo-controlled trial	160	100	59.3	Postmenopausal women	Intervention group (n = 80) and control group (n = 80)	9 m	9 m	1000 IU		↑ 25(OH)D ↓ PTH	↓24.2% in s-CTX, 13.4% in P1NP, and 21.3%		-
[32]	USA	Prospective, randomized, double-blind, placebo-controlled trial	258	100	68.2	Healthy African American women with serum 25(OH)D 20–65 nmol/L	VitaminD3 vs. placebo	3 y	Every 6 months	Adapted dose to achieve a concentration of 30 mg 25(OH)D in the serum >75 nmol/L	↑BMD of spinal cord ns change in total BMD	↑25(OH)D	NR	ns changes in risk of falling	-
[33]	Denmark	Double-blinded placebo-controlled randomized trial	81	100	60–79	Healthy postmenopausal women with 25(OH)D < 50 nmol/L and PTH> 6.9 pmol/L	Intervention group (n = 40) and control group (n = 41)	3 m	3 m	70 µg (2800 IU) Daily	↑ at the trochanter and femoral neck	↑25(OH)D ↑1,25(OH)2D ↓PTH	ns changes in BSAP P1NP Osteocalcin CTx	NR	↓failure load ↑trapezoidal thickness and ↑estimated bone strength at the tibia
[34]	Brazil	Double-blind, placebo-controlled trial	160	100	58.8	Individuals with BMD> −1.5 SD	Intervention group (n = 80) and control group (n = 81)	9 m	9 m	1000 IU Daily	NR	↑25(OH)D in intervention group ↓ in the control group ns change in PTH	NR	↑ rate of falls (OR: 1.95 95% CI, 1.23–3.08) and recurrent falls (OR 2.8, 95% CI, 1.43–5.50)	
[35]	USA	Prospective, randomized, double-blind, placebo-controlled trial	260	100	68.2	Postmenopausal women	Intervention group (n = 130) and control group (n = 130)	3 y	Annually	Adapted dose to achieve a concentration of 75–172 nmol/L (doses of 60, 90, and 120 mg)	↓ Femoral neck BMD in all groups	↑25(OH)D		NR	-
[36]	USA	Double-blind, placebo-controlled randomized clinical trial	218	100	59.6	Postmenopausal women	Intervention group (n = 109) and control group (n = 109)	12 m	12 m	2000 IU Daily (+ weight loss diet)	ns changes in spine and femoral neck BMD	↑25(OH)D			ns change in upper body muscle strength ↓leg strength in the vitamin group D compared to placebo
[37]	Finland	Double-blind, placebo-controlled trial	350	100	74	Elderly women	Group a (intervention): n = 102; Group b intervention): n = 103 Group c (intervention):n = 102, and Group d (control): n = 102	1 y	1y, 2 y assessment	20 μg (800 IU) +/- exercise Daily	↓ Femoral neck BMD in all groups	↑25(OH)D		ns change in falls	
[38]	Austria	Single-center, double-blind, randomized placebo-controlled trial	192	0	43	Healthy men	Intervention group (n = 100) and control group (n = 100)	3 m	3 m	20,000 IU Weekly	↓ femoral neck BMD in men with baseline 25(OH)D levels ≥ 50 nmol/L (n = 115) ns changes in total body BMD, lumbar spine BMD, hip BMD	↑25(OH)D and ns changes in PTH in subjects with 25(OH)D levels < 40 nmol/L ↑25(OH)D and ↓PTH in subjects with 25(OH)D levels > 40 nmol/L	ns changes in CTX, OC		ns changes in BTM, TBS
[40]	Great Britain	Single-center, parallel-group, participant-randomized, double-blind interventional trial	379	48	75	Patients in lack of treatment for osteoporosis, hyperparathyroidism, history of fractures, hypercalcemia, hypocalcemia	3 intervention groups (~110 per group)	1 y	1 y	300 μg 600 μg 1200 μg (12.000, 24.000 and 48.000 IU) Monthly	ns change in bone density	↑25(OH)D	NR	ns changes in falls	-
[41]	Canada	Double-blind, randomized clinical trial	311	47	62.2	lumbar spine and total ischial BMD T score >−2.5 SD, serum 25(OH)D: 30–125 nmol/L and normal serum Ca 2.10–2.55 mmol/L	3 parallel groups and control group (n~100 per group, total n = 400)	3 y	DXA: 12, 24 and 36 months (HR-pQCT: 6, 12, 24 and 36 months	400 IU 4000 IU 10,000 IU Daily	↓ radial BMD at 4000 IU/day or 10,000 IU/day ↓tibial BMD at 10,000 IU per day	↑25(OH)D ↓PTH	↑CTx	ns change in falls	ns changes in failure load ns differences in bone strength in either the stapes or the tibia
[42]	Canada	Randomized clinical trial	311	47	62.2	total hip BMD total hip T score >−2.5 SD, serum 25(OH)D between 30 and 125 nmol/L and serum Ca 2.10–2.55 mmol/L	3 parallel groups (n~100 per group)	3 y	DXA: 12, 24, and 36 months (HR-pQCT: 6, 12, 24, and 36 months	400 IU 4000 IU 10,000 IU Daily	↓ BMD in women but not men ↓1.8% (400 IU), 3.8% (4000 IU) and 5.5% (10,000 IU) at the radius. Men ↓ 0.9% (400 IU), 1.3% (4000 IU), and 1.9% (10,000 IU) at the radius. In the tibia, losses in tBMD were smaller but followed a similar trend	↑25(OH)D ns PTH	ns CTx	NR	ns bone strength changes
[43]	Iran	Single blind Clinical trial	400	48.5	20–60	Healthy adults	Vitamin D (n = 76)	8 w	8 w	50,000 Weekly	↓ osteoporosis in the intervention group				-
[44]	Shanghai	Randomized, double-blind, Placebo-controlled trial	448	69	31.9	Vitamin D-deficient adults’ serum 25(OH)D: 12.5–50 nmol/L	Placebo (n = 222) Intervention group (n = 226)	20 w	20 w	2000 IU Daily		↑25(OH)D	↑bALP ns change in serum PINP, β-CTX, or TRAP5b In intervention group, subjects with 25(OH)D ≥75 nmol/L ↑β-CTX and TRAP5b, but smaller ↓ in Ca and Ca product phosphorus		
[46]	Austria	Randomized, double-blind, placebo-controlled trial	289	37.5% control g and 35.3 vitamin D	62.2 control and 60.3 vitamin D	Patients in ICU	Placebo (n =136) Vitamin D (n = 153)	6 m	6 m	Initial: 540,000 IU 90,000 IU Monthly	ns change in BMD at the lumbar spine and femoral neck	↑25(OH)D ↓ PTH	ns changes in CTX and OC	ns changes in falls	-
[47]	New Zealand	Randomized, double-blind, placebo-controlled trial	452	35% control and 38% vitamin d	69	Adults living in the community	Placebo (n = 224) Vitamin D (n = 228)	2y	2y	100,000 IU (2.5 mg) Monthly First dose was double	ns change in lumbar spine BMD ↓ proximal femur and total body BMD in all groups	↑25(OH)D			
[48]	India	Controlled trial	16	0	18–35	Men with vitamin D deficiency	Intervention group (n = 8 and a control group (n = 4)	3 y	3 y	60,000 IU Weekly		↑25(OH)D			↑ Bone mineral balance
[49]	Austria	Single-center, double-blind, placebo-controlled, parallel-group study	197	47	62.4	People with arterial hypertension and serum 25(OH)D concentration <75 nmol/L	Vitamin D (n = 98) Placebo (n = 99)	8 w	8 y	2800 IU			ns changes in bALP, CTX, OC and P1NP values ↑ OC in men.		-

25(OH)D = 25-hydroxyvitamin D; BMD = bone mineral density; PTH = parathormone; vBMD = volumetric bone mineral density; HR-pQCT = high-resolution peripheral quantitative computed tomography; TtBMD= total volumetric bone mineral density; BTM= bone turnover markers; TBS = trabecular bone score; lbs = pounds; s-CTX = serum carboxy-terminal collagen crosslinks; P1NP = propeptide of type 1 procollagen; CI = confidence interval; ICU = Intensive care unit; bALP = bone alkaline phosphatase; CTX = C-terminal telopeptide; OC = osteocalcin; β-CTX = β-isomerized C-terminal telopeptides; TRAP5b = Tartrate-resistant acid phosphatase; y = year; m = month; w = week; WM = women; ns = non-significant; OR = odds ratio.

Bone turnover markers appear to be associated with factors other than vitamin D supplementation. Interestingly, body mass index (BMI) may play a role in the concentration of these markers, since individuals with BMI < 26.4 kg/m2 (n = 100) had significantly higher values of CTX and OC [38]. Furthermore, the levels of vitamin D in the blood serum may also play an important role. For example, in vitamin D deficiency, bone alkaline phosphatase (ALP) is considered as a good index, since its increment is related to rickets [50] and osteomalacia [51]. Among the participants treated with cholecalciferol, subjects who achieved serum 25(OH)D ≥ 75 nmol/L had greater increases in serum β-CTX (224% vs. 146%, *p* = 0.02) and TRAP5b (22.2% vs. 9.1%, *p* = 0.007), but smaller decreases in serum calcium (−1.3% vs. −1. 9%; *p* = 0.005) and calcium–phosphorus product (−2.6% vs. −3.3%; *p* = 0.02), compared with subjects with serum 25(OH)D < 75 nmol/L [44].

### 3.2. Combined Vitamin D and Ca Supplementation

In total, nine studies fulfilling the inclusion criteria were identified (total n = 1568 subjects) [21,24,25,26,27,28,29,30,52]. The studies were conducted in several countries, i.e., USA [28,29,30,52], Lebanon [26,52], Turkey [25], Australia [21], India [27], and China [24]. The mean age of participants ranged from 32 to 71 y, and most of them included only women [25,28,29,30,52] or mostly women [24]. The minimum treatment duration was 16 weeks [21], some studies had a 6 month intervention period [25,29], and most studies had a 12 month [24,26,27,28,30,52] intervention period. The dose of vitamin D ranged from 400 IU per day to 4800 IU per day, and the dose of administered dietary and supplementary calcium ranged from 200 mg to 1500 mg per day. The results are summarized in Table 2, and they are presented below in detail.

#### 3.2.1. Combined Vitamin D, Ca Supplementation, and BMD

All [21,24,25,26,27,29,30,52] but one study [28] measured BMD before and after the intervention. Increases in total BMD were observed in several studies [26,27,29,52], which were more significant in high-dose supplemented groups [26,52]. More specifically, 3750 IU vitamin D/day and 1000 mg Ca/day were compared to 600 IU vitamin D/day and 1000 Ca/day, and the high dose was more effective in increasing BMD [26,52]. However, some studies showed no effect of vitamin D and calcium supplementation on total BMD [21,25,30], even in high doses of 4800 IU vitamin D per day and 1200 mg Ca per day [30]. Site-specific BMD was also assessed. Increases were observed in lumbar spine BMD [24,25,26,27], while in some studies, no difference in lumbar spine was documented after vitamin D and Ca intervention [30,52]. Hip BMD was increased in one study [26], and remained unchanged in the rest of the studies [24,30,52]. It is noted that in the study of Rahme et al., there were significant increases in the percentage change in total hip and lumbar spine BMD, but not in the femoral neck at 12 months of intervention within groups, while the percentage change in the subset of BMD increased significantly only in the high-dose group [26]. Interestingly, in the study of Pop et al., a differentiation of results was documented, depending on the analysis followed [28]. Specifically, in the as-treated analysis, no differences were observed between BMD in the vitamin D3-treated groups for femoral neck, rostrum, lumbar spine, total hip, and total-body BMD, in the as-treated analysis [28]. On the other hand, the intention-to-treat (ITT) analysis yielded a statistically significant interaction between treatment and time for cortical thickness (*p* < 0.03), while there was a loss of total and trabecular IP over time in all groups (*p* < 0.05) [28]. Both ITT and treatment analyses showed a decrease over time in trabecular volumetric BMD in all women (*p* < 0.05), but between groups, the changes were not significant [28]. These differences were not associated with higher levels of vitamin D3 intake (2000 and 4000 IU) [28]. Furthermore, no ethnic difference has been found in the percentage change in BMD (African American vs. Caucasian) [30], or in the correlation between the absolute change in serum 1,25(OH)2D level and the change in BMD [30].

The observed differences may be differentiated according to baseline serum 25(OH)D and/or PTH levels. Indeed, there was a significant difference in total body BMD by dose, in subjects with baseline serum 25OHD ≤50 nmol/L, while the percentage change in BMD was not significant [30]. It is noted that in the adjusted analysis for multiple comparisons, the pairwise differences were not significant [30]. Subjects with serum 25(OH)D concentrations of below 20 ng/mL, and a PTH level of above 76 pg/mL showed a trend for a higher percentage change in BMD at 12 months in all skeletal regions in the high-dose group compared to the low dose group, which reached significance only in the total hip at 12 months [26].

Correlations have been documented between BMD and other factors, such as free- and total 25(OH)D and certain biomarkers. An association of 25(OH)D, but not the free or bioavailable form, and the percentage change in thigh neck BMD was observed (y = 0.059X − 1.28, R^2^ = 0.021, *p* = 0.033), whereas bioavailable (y = 0.353X − 0.729, R^2^ = 0.022, *p* = 0.029) and free (y = 0.154X − 0.834, R^2^ = 0.025, *p* = 0.018), but not total 25(OH)D were associated with total body BMD [30]. In addition, serum homocysteine levels were inversely correlated with the BMD of femoral neck (r = −0.49, *p* = 0.009), trochanter (r = −0.5, *p* = 0.009), Ward’s angle (r = −0.45, *p* = 0.022), and total hip (r = −0.43, *p* = 0.002) [30]. Furthermore, for spinal BMD, there was a small linear increase with increasing serum 25(OH)D, but no significant correlation with femoral neck, total body, or serum NTx [30].

**Table 2 diseases-11-00029-t002:** Characteristics of included randomized controlled trials with Vitamin D and Calcium supplementation.

Authors	Country	Study Design	n	Sex (WM %)	Age (Mean)	Sample (at Baseline)	Groups	Duration of Intervention	Follow-Up	Dose of Vitamin D, Frequency	Effects on BMD	Effects on 25(OH)D and PTH	Effects on Bone Turnover Indices	Effects on Falls	Secondary Results
[21]	Australia	Randomized placebo-controlled doubled-blind RCT	54	n/a	32	Overweight or obese vitamin D-deficient (25OHD < 50 nmol/L) adults	Vitamin D n = 28 placebo group n = 26	16 weeks	16 weeks	Loading dose of 100.000 IU cholecalciferol, followed by 4000 IU cholecalciferol/d or a matching placebo	ns changes in BMD	↑25(OH)D ↓ PTH	ns changes in FGF-23	ns change in OF	
[24]	China	randomized controlled trial	420	81.4	>60	bone mineral density (BMD) at lumbar vertebra or hip ≤ −2.5	Inactive vitamin group (n = 98) Inactive vitamin with exercise group (n = 97) Active vitamin group (n = 99) Active vitamin and exercise group (n = 98)	12 m	3,6,12 m 12 m BMD	Inactive VitD group: 800 mg Ca and 800 IU inactive VitD/day. Inactive VitD + exercise group: 800 mg Ca and 800 IU VitD/day + instructions to improve muscle strength and balance Active VitD group: 800 mg Ca and 0.5 µg active VitD/day Active VitD + exercise group: 800 mg Ca and 0.5 µg of active VitD/day + instructions for improving muscle strength and balance	↑Lumbar BMD of the A VitD group and the P-A VitD group Ns change in hip, femur neck BMD	↑25(OH)D		ns change in OF ns change in falls	-
[25]	Turkey	Prospective, open-label, controlled clinical trial.	120	100	50	Pre- and postmenopausal women diagnosed with vitamin D deficiency	Group A (cholecalciferol + Ca) n = 43 Group B (calcitriol + cholecalciferol + Ca) n = 77	6 m	6 m	Group A (1000 IU of Vitamin D3 and 1 g of Ca/D) Group B (0.5 μg calcitriol in addition to 400 IU of cholecalciferol and 1 g of Ca/D)	Ns in total BMD ↑ Lumbar spine BMD in group B	↑25(OH)D ↓PTH	↓ ALP (Group B) ↑ CTx, NTx, deoxypyridinoline, OC (Group A and Group B, no difference between groups)	ns change in OF	-
[26]	USA	Randomized double-blind, controlled trial	222	99	71	Elderly (>65 years), overweight with a serum 25(OH)D between 10–30 ng/m	High dose group (n = 110) Low dose group (n = 112)	6, 12 m	6,12 m	Supplementation with 1000 mg of elemental Ca citrate/day, and the daily equivalent of 3750 IU/day or 600 IU/day of vitamin D3	↑BMD at the total hip and lumbar spine, but not the femoral neck, in both study arms. ↑ subtotal body BMD in the high-dose group at 1 year. Subjects with 25OHD < 20 ng/mL and PTH level > 76 pg/mL ↑ hip BMD	↑25(OH)D ↑ calcitriol in the high dose group ↓ PTH but ns change between groups	↓OC, CTX ns difference between groups	↑ in OF	
[27]	India	Randomized, open-labeled, comparative, controlled clinical study	65	66	40	Osteopenic adults	Treatment group: 32 Control group: 33	0, 6, 12m	6, 12 m	Treatment group received two tablespoons of PG (10mL in lukewarm milk), along with Ca and vitamin D3 supplements (containing elemental Ca 1200 mg and vitamin D3 [cholecalciferol] 800 IU/day) twice a day, whereas control group received only Ca and vitamin D3 supplements twice a day	↑BMD scores at 6 months, which was sustained at 12 months in both the study groups. Maximal improvement was observed in the lumbar spine and left forearm regions.	↑vitamin D3 in the PG group than in the SOC group at 6 and 12 months, which was statistically significant at 12 months (30.3 ng/mL vs. 22.3 ng/mL)	Improvement in OC, TRAP-5b in the PG-treated group	ns change in OF	
[28]	USA	Randomized, double-blind controlled study	58	100	58	overweight/obese healthy, postmenopausal women (age 50–70 years old; BMI 25–40 kg/m2)	A: 600 IU/day (n = 19), B: 2000 IU/day (n = 20), C:4000 IU/day (n = 19)	12 m	12 m	Vitamin D 600, 2000, 4000 IU Ca 1.2 g/day during weight control	↓ cortical thickness in the 600-IU group but not in the higher vitamin D groups	↑25(OH)D ↓ PTH	↑ CTX, P1NP ns difference between groups	ns change in OF	3 % weight reduction
[29]	USA	Randomized trial	135	100	55.8	Overweight/obese Caucasian, early–postmenopausal women	Placebo n = 62 Dairy n = 64 Supplement (Ca + vitamin D) n = 62	6 m	6 m	Moderate energy restriction (~85% of energy needs) for all participants. All subjects complemented with low-fat dairy foods (4–5 servings/day), or Ca + vitamin D supplements a total of ~1500 mg/day and 600 IU/day of Ca and vitamin D, respectively, or placebo pills	Supplement group: lower decrease or slight increase in BMD in measured skeletal sites.	↑25(OH)D ↓ PTH	ns change in OF	ns change in OC, NTx ↓ Urinary CTx in the supplement group and ↑ in the control group	dairy group: better body composition outcomes, higher decrease in fat and lower decrease in lean mass.
[30]	USA	Randomized placebo controlled trial	273 (Caucasian n = 163 African American n = 110)	100	Caucasian 67 African American 65	Elderly women with vitamin D insufficiency, (serum 25(OH)D levels ≤50 nmol/ L)	8 intervention groups D3 doses of: 400 IU/d, n = 20 800 IU/d, n = 22 1600 IU/d, n = 23 2400 IU/d, n = 24 3200 IU/d, n = 21 4000 IU/d, n = 20 4800 IU/d, n = 21 Placebo group n = 22	12 m	12 m	Vitamin D3 400, 800, 1600, 2400, 3200, 4000, or 4800 IU daily Ca 200 mg as to maintain a total Ca intake of ~1200 mg	ns change in total BMD and hip, lumbar spine BMD No association between change in BMD and the 12-month values for serum total 25(OH)D, serum free 25(OH)D or serum 1,25(OH)2D	↑25(OH)D ↓PTH		ns change in OF	Results for Caucasian and African American women were similar. ↑ in total body Ca in the treated women with higher baseline serum PTH.
[51]	USA and Lebanon	Double-blind, randomized controlled trial	221	55.2	>65 71.1	Overweight, with a baseline serum 25(OH)D of between 10 and 30 ng/mL	High-dose group: 1000 mg elemental Ca and 3750 IU/day vitamin D. Low-dose group: 1000 mg elemental Ca and 600 IU/day vitamin D	6, 12 m	6, 12 m	All subjects received 1000 mg elemental Ca and oral vitamin D3 (600 IU/ day or 3750 IU/day) supplementation	ns change in spine and hip BMD at 12 months ↑ subtotal body BMD with the high dose.	No increase in total, bioavailable, and free 25(OH)D levels was found at 12 months (*p* < 0.001) for low dose and high-dose supplementation. Vitamin D supplementation at a dose of 3750 IU/day resulted in serum levels of total, bioavailable, and free 25(OH)D, that were 1.28–1.38 higher than levels reached with 600 IU/day dose.			Weak but significant relationship between 25(OH)D and % BMD change at femoral neck only (*p* = 0.033), and only mild significant correlation between the free and bioavailable 25(OH)D, and the total body BMD at 12 months.

Ca = calcium; BMD = bone mineral density; PTH = parathormone; OC = osteocalcin; CTX = C-terminal telopeptide; NTx: N-terminal telopeptide; TRAP-5b = Tartrate-resistant acid phosphatase; FGF-23 = Fibroblast growth factor 23; y = year; m = month; w = week; WM: women.

#### 3.2.2. Combined Vitamin D and Ca Supplementation, and Circulating 25(OH)D

All RCTs evaluated the change in 25(OH)D in blood serum and documented increases either within the intervention groups, or between the intervention and control groups [21,24,25,26,27,28,29,30,52]. In a one-year intervention, subjects receiving the highest dose of vitamin D (3750 IU) +1200 mg Ca had a change in serum levels from 20.65 ± 7.89 to 36 ± 9.73 ng/mL compared to subjects supplemented with a low dose of vitamin D (600 IU) + 1200 mg Ca, in which levels changed from 20.06 ± 6.92 ng/mL to 25.96 ± 6.88 ng/mL (*p* < 0.001) [30]. In addition, free 25(OH)D increased from 5.33 ± 1.87 to 6.86 ± 2.07 ng/L after a low dose, and from 5.38 ± 2.14 to 8.92 ± 3.15 ng/L after high dose supplementation [30]. In a 6-month intervention in two groups (one low dose 400 IU vitamin D +0.5 μg calcitriol+ 1000 mg Ca and one high-dose 1000 IU vitamin D+ 1000 mg Ca) per protocol analysis gave the following results: after 6 months of treatment, serum 25(OH)D levels increased from 11.8 ± 8.8 to 59 ± 35 ng/mL in the cholecalciferol and Ca group, and from 8.7 ± 6.2 to 28 ± 29 ng/mL in the calcitriol, cholecalciferol, and Ca groups, and a significant difference was also observed between the groups (*p* = 0.05) [25]. In another intervention, only the group receiving inactive vitamin D and the group receiving inactive vitamin D along with exercise had a significant increase at 12 months [24]. In obese patients, the increase in serum 25(OH)D in response to 600, 2000, or 4000 IU vitamin D3/day was 3.8 ± 4.1, 7.4 ± 6.5 and 14.1 ± 8.1 ng/mL, respectively, with the interaction between time and treatment being significant.

#### 3.2.3. Combined Vitamin D and Ca Supplementation, and Circulating Calcium

Differentiated effects were found in serum calcium concentrations in blood serum after the intervention. One RCT identified no significant differences in serum calcium concentrations between groups at baseline or after intervention, and no subjects had hypercalcemia (>10.5 mg/dL) at 12 months [28]. Another study observed a decrease in mean serum calcium concentrations after 16 weeks within the placebo group, but with no differences in mean calcium change compared to the vitamin D group [21]. Feng et al. showed that there was a significant increase in blood calcium levels in all groups [24], while Tenekol et al. documented increases only in the group receiving calcitriol, cholecalciferol, and Ca [25]. Finally, Munshi et al. recorded an increase in serum calcium—although this was not statistically significant—in the supplementation group, while a decrease was observed in the placebo group at the end of 6 and 12 months [27].

#### 3.2.4. Combined Vitamin D, Ca Supplementation, and PTH

Six studies [21,25,26,28,29,30] assessed PTH levels in blood serum, and all documented a decrease in the intervention group. After 16 weeks of vitamin D supplementation, a decrease in the mean iPTH concentrations was observed within and between the two groups, with the difference only within the treatment group being significant (4.89 ± 1.86 pmol/L vs. 3.71 ± 1.31 pmol/L, *p* = 0.002) [21]. Similarly, the mean serum PTH level decreased from 131 ± 62 to 71 ± 51 pg/mL in the cholecalciferol and Ca group, and from 235 ± 193 to 58 ± 38 pg/mL in the calcitriol, cholecalciferol, and Ca groups (*p* = 0.003, between groups) [25]. In addition, serum PTH was significantly reduced only in participants in the supplementation group, and not between the three groups (placebo, dairy, vitamin D, and calcium supplements) [29]. Pop et al., after 12 months of treatment, concluded that PTH decreased by 10.6 ± 16.6% (*p* < 0.01); however, the change did not differ significantly between groups of different vitamin D dosing (500 IU, 2000 IU, and 4000 IU) [28], while Rahme et al. (2017) recorded a gradual decrease in PTH levels at 6 and 12 months, only in the low supplemental dose group (600 IU) vs. a higher dose (3750 IU) [26]. In terms of ethnic groups, serum PTH decreased as circulating vitamin D was increasing, in both Caucasian and African American women [30]. There was no correlation between baseline serum PTH level and the percentage change in spine BMD, femoral neck BMD, or serum NTx, but there was a significant positive correlation between the PTH level and the percentage change in total body BMD (r = 0.18, *p* = 0.020) [30].

#### 3.2.5. Combined Vitamin D and Ca Supplementation, and Falls/Fractures

One study recorded a significant increase in OF [26], while the others observed no difference between the groups in the OF [21,24,25,27,28,29,30]. Only one study with combined vitamin D and Ca supplementation included in the review evaluated falls in participants, but the differences were not significant [24].

#### 3.2.6. Combined Vitamin D and Ca Supplementation, and Bone Turnover Biomarkers

A total of six studies measured the concentration of bone turnover markers in the plasma of the participants [21,25,26,27,28,29]. Two showed an increase in bone metabolism indicators [25,28], while three observed a decrease in concentration [26,27,29], and one reported non-significant changes [21].

More particularly, OC increased by 43 ± 104% in the high-dose vitamin D group, and by 111 ± 276% in the low dose vitamin D group (*p* > 0.05 between groups) [25]. In contrast to these findings, Rahme et al. observed a 20% to 22% decrease in the OC and CTx levels in both groups (low and high dose), with no significant differences between the two groups [26]. Decreases in both the OC and serum TRAP-5b levels, with the decline being significantly higher in the treatment group compared to the standard care group, were documented, indicating reduced bone turnover [27]. As far as CTx is concerned, in the study of Ilich et al., it increased in the control group and decreased in the vitamin D+ Ca supplementation group [29], while in the study of Tanakol, both CTx and NTx increased [25]. PGF-23 was measured in only one study, and no differences were recorded after vitamin D + Ca supplementation [21].

## 4. Discussion

A total of 26 randomized clinical trials were included in this review to summarize the recent data from vitamin D and calcium supplementation, both solely and in combination, and to study their effects on bone density, circulating vitamin D, calcium, parathyroid hormone, markers of bone turnover, and clinical outcomes, i.e., falls and osteoporotic fractures.

The ingested vitamin D is hydroxylated in the liver and kidneys, and is converted to its active form, i.e., 1,25-dihydroxyvitamin D (1,25(OH)2D) [53]. This active circulating form of the vitamin binds to vitamin D receptor (VDR), exerts several actions, and regulates gene expression [53]. It increases intestinal calcium and phosphorus absorption, it stimulates calcium renal reabsorption, and it interacts with PTH and activates osteoclasts that are responsible for bone resorption [20]. In this context, vitamin D promotes calcium absorption and helps to maintain adequate serum calcium concentrations to enable the normal mineralization of the bone. In parallel, vitamin D is needed for bone growth and bone remodeling by osteoblasts and osteoclasts [54].

Consistently with clinical studies on bone density, only few studies detected significant differences in BMD after sole vitamin D supplementation [26,39,43], which is in line with a meta-analysis regarding vitamin D effects [55]. The co-administration of calcium and vitamin D led to increases in total BMD in most studies [26,27,29,52], which were more significant in high-dose supplemented groups [26,52], while in others, it had no effect [21,25,30]. A meta-analysis of 23 trials prescribing vitamin D plus calcium (19 double-blinded trials, and 18 placebo controlled trials) with a mean duration of 23.5 months, including 4083 participants (92% women, mean age 59 years) with variable ingested vitamin D doses (<800 IU/day in 10 trials), reported a small increase in BMD at the femoral neck, but not at other sites, that the authors attributed to chance [47]. A high dose- and long-term supplementation of vitamin D may be also associated with bone loss, probably due to a decrease in plasma PTH [41], which is observed in both men [39] and women [18]. Changes in PTH with vitamin D supplementation are likely to directly affect parathyroid cells through an effect on osteoclast (CTx) activity leading to a dose-dependent accelerated decrease in bone formation and observed volumetric BMD [33,41].

Moreover, several factors may affect the effectiveness of vitamin D intervention, such as sex, baseline BMD, baseline vitamin D status, and the duration of the treatment. For example, Burt et al. reported that during the menopausal and postmenopausal period, a high dose of sole vitamin D supplementation is required in women closer to age 55 and after age 65, due to greater bone loss and accelerated bone loss, respectively [41]. A clinical trial at the bone density clinic at Kowsar Hospital in Semnan showed that the prevalence of vitamin D deficiency in osteopenic and osteoporotic patients was higher than in normal subjects, an observation which was associated with age and sex [43]. Furthermore, it was shown that the incidence of osteoporosis decreased through vitamin D treatment [43]. In these studies, the duration of the treatment was ~3 years [26,39,43], while studies showing no effect had a duration of ~1 y or less [36,37,38,40] suggesting a significant role of treatment duration to achieve a beneficial effect on BMD. However, a longer duration of treatment in healthy subjects may not have a positive change on BMD [48], while lessened or null effects have been observed in men [38]. Serum 25(OH)D has been positively associated with BMD [56], but the effects of baseline 25(OH)D on the effectiveness of supplementation are less clear. Some studies show no effect of baseline serum 25(OH)D on the outcomes of supplementation [30], while a potential effect in total hip BMD may be present, if low serum 25(OH)D is combined with high PTH (>76 pg/mL) [26]. Moreover, a beneficial effect was observed in subjects with low baseline 25 OHD levels, and at doses < 800, as compared to > 800 IU/day [47].

All [13,18,33,34,36,37,38,39,40,46,48] but one [35] study included in this narrative review showed that circulating 25(OH)D levels increased after sole vitamin D supplementation, even in cases of low baseline 25(OH)D concentration [34]. Similarly, all included RCTs reported an increase in circulating 25(OH)D after supplementation with both vitamin D and calcium [21,24,25,26,28,29,52], although according to a meta-analyses, circulating 1,25(OH)2D may be suppressed through calcium co-administration (vitamin D alone increased circulating 25(OH)D by 18.6 pmol/L; 95% CI = 12.7–24.4 pmol/L, while vitamin D plus calcium increased circulating 25(OH)D by 4.9 pmol/L; 95% CI = 0.4 −10.2 pmol/L; *p* 0.001) [57]. In addition, a dose-dependent and significant increase in serum 25(OH)D was observed after the intervention with sole vitamin D [21,24,25], or its combination with calcium [41]. Our results are in line with the findings of a meta-analysis in overweight and obese adults that concluded that a daily dose of 1000 IU vitamin D supplement can increase serum 25(OH)D levels and reduce PTH levels [58]. Along with the interpretation of the results regarding circulating vitamin D, it should be also noted that higher plasma 25(OH)D levels were observed during the period of July–September, compared to the period of January–March [38].

Differentiated results were present in the case of circulating calcium pre- and post- supplementation, since some studies reported a non-significant effect [21,27,28] or an increase [24] after the intervention with vitamin D and calcium, or with a combination of calcitriol, vitamin D, and calcium [25].

All studies that measured serum PTH pre- and post- intervention with vitamin D reported a significant decrease in supplementation groups [21,25,26,28,29,30]. The percentage of PTH decrease was dependent on the duration, dose, and ethnicity [26,30]. Indeed, in Caucasian and African American women, serum PTH decreased as circulating 25(OH)D increased over the range of serum 25(OH)D 50–175 nmol/L, suggesting a vitamin D–PTH interaction [59,60]. Indeed, vitamin D reduces PTH gene expression, and it has an inhibitory effect on the proliferation of parathyroid cells [20]. However, the combined effects of calcium and vitamin D supplementation on PTH are less clear-cut. Evidence from RCTs suggests that bone benefits are evident with vitamin D supplementation at adequate calcium intakes and at normal levels of PTH [61]. In a previous review, vitamin D was found to lower PTH, but this effect was not apparent in the presence of calcium supplementation [62]. Interestingly, Lerchbaum et al. (2019), reported that vitamin D supplementation reduced PTH levels only in normal-weight men, whereas in men with higher BMIs, there was no significant effect [38]. It was thus hypothesized that overweight/obese subjects might require higher vitamin D doses in order for beneficial effects to be exerted on PTH levels [38]. It is well known that BMI has a unidirectional negative effect on vitamin D metabolism. Fillipo et al. has also previously described the possible intrinsic beneficial metabolic effects of vitamin D itself in overweight and obese patients, which influence adiposity and metabolic outcomes [63].

It is not clear whether the improvement in circulating vitamin D through supplementation can reduce the frequency of falls and fractures. Indeed, most studies reported no significant effects of vitamin D [37,40,41,48] or combinations of vitamin D with calcium [24] on falls. Other studies have reported that vitamin D and calcium supplementation (not only calcium) leads to an increase in BMD and reduces the risk of OF [12,31,64,65], while others have shown no effect, supporting our results [66,67]. Our results are similar to a meta-analysis that concluded that vitamin D had no effect on the total fractures, hip fractures, or falls [55], while several meta-analysis have shown that the combination of calcium and vitamin D significantly reduced the fractures risk [12,65,68]. This reduction was more pronounced among women with low plasma 25(OH)D concentrations and calcium co-administration [34]. Interestingly, it has been noted that an intermittent high dose supplementation of vitamin D was associated with an increased risk of falls and fractures [69]. The exoskeletal effects of vitamin D are strongly correlated with the development of sarcopenia disease [63]. Moreover, it is possible that the total diet and/or dietary protein may be important in addressing this issue, since sarcopenia often co-exists with osteoporosis (osteosarcopenia) [70]. In this case, animal protein may be a key factor for modifying body composition and sarcopenia, such as in the case of cancer, which is also related to sarcopenia [71,72]. In addition, inflammatory indices, such as platelet activating factor, are related to OF [73], and an antioxidant-rich diet and/or fatty acid profile may lessen its deleterious actions [74,75].

Regarding the effects of vitamin D supplementation on bone turnover markers, such as CTx, bALP, OC, TRAP-5b, and P1NP, most studies included in the present review found a no significant effect of vitamin D [18,35,42,44,46,48,49]. Interventions with supplements of vitamin D and calcium have shown increases [25,28], decreases [26,27,29], or no significant changes [21]. A significant decrease in the serum levels of bone turnover markers may possibly result after large doses of calcium (1000 mg elemental) taken in combination with vitamin D [26], or in a combination of calcium and vitamin D supplements with other substances, such as collagen peptides [76]. The effects of vitamin D on bone turnover markers may be limited to subjects with low 25(OH)D levels [77],while no effects on CTx and OC in men with low 25(OHD levels (<50 nmol/L, n = 85) were documented [38]. The effects of vitamin D on bone turnover may also depend on the body’s calcium status, since in cases of hypocalcemia, active vitamin D may also activate osteoclasts to achieve calcium homeostasis [54]. In general, these small effects of vitamin D supplementation on bone turnover biomarkers are in line with previous studies performed in hypertensive patients [49], healthy postmenopausal women [33], and healthy young and elderly adults [78], as well as healthy obese men and women [79]. Moreover, several inter-correlations between them may affect the observed results [49,80,81], while glucose and HbA1c may also affect their levels [82,83].

Besides environmental factors, the genetic background may also influence the effects of supplementation [71]. For example, 35 genes and several single-nucleotide polymorphisms (SNPs) have been connected to vitamin D status, which could potentially alter individual 25(OH)D responses to supplementation [84]. In parallel, dietary habits may influence vitamin D status [71,85],while gene–diet interactions may also take place [86], although they may not at last largely affect BMD [87,88]. In addition, concomitant calcium supplementation may reduce compliance with vitamin D, due to calcium-induced digestive problems, which probably supported the decision in some trials to try vitamin D supplementation alone [89]. Increased vitamin D doses may also be related to hypercalcemia and hypercalciuria [41,90], with the risk being higher in subjects with calcium co-administration [32]. However, not all studies documented a high risk of hypercalcemia [36,40].

Limitations of several studies were the short duration, the low sample size, [38,40,46], and no placebo comparator during the study period [40], while other factors such as exercise, muscle, and weight loss may affect BMD more than vitamin D alone [46,70]. Gender bias may be also inherent, since most studies have been conducted in women, and the age distribution is quite narrow and shifted to older ages [91]. Moreover, no much data on ethnic-related differences are reported [91]. Last but not least, several vitamin D–drug interactions should be taken into account. Statins increase vitamin D [92] and anti-epileptic treatment, and are often used as co-analgesic treatment in the elderly [93], and they induce vitamin D catabolism [94]. In real life, other medicines for osteoporosis may be prescribed [14], and potential additional diet–drug interactions should be taken into account.

## 5. Conclusions

Osteoporosis is a systemic skeletal disorder with significant negative effects on the general health and quality of life after menopause and in old age. In addition to specific pharmacological and/or hormonal treatments, supplementation strategies seem to be very important, and they are generally recommended [14]. Vitamin D supplementation, alone or in combination with Ca, is considered as being fundamental to enhancing the positive effects of any particular therapy in patients, who are more fragile and at higher risk of vertebral and non-vertebral fragility fractures due to disorders related to bone metabolism (e.g., osteoporosis or vitamin D deficiency). The present reviewed evidence suggests that vitamin D alone or in combination with calcium increases circulating 25(OH)D. Calcium with concomitant vitamin D supplementation, but not vitamin D alone, leads to an increase in BMD, while no significant differences were documented for the reduction in the risk of total fractures. Figure 1 presents a summary of the conclusions of the review. Further research is needed to investigate the possible role of vitamin D and calcium on bone turnover markers.

## Figures and Tables

**Figure 1 diseases-11-00029-f001:**
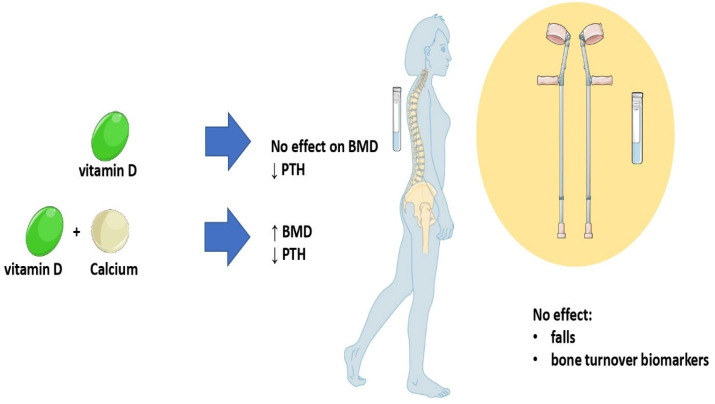
Summary of the main conclusions. Part of the figure were drawn by using pictures from Servier Medical Art. Servier Medical Art by Servier is licensed under a Creative Commons Attribution 3.0 Unported license (http://creativecommons.org/licenses/by/3.0/ (accessed on 12 January 2023)).

## Data Availability

Not applicable.

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
