# Peer review of "Vitamin D and Calcium in Osteoporosis, and the Role of Bone Turnover Markers: A Narrative Review of Recent Data from RCTs"

_diseases, 2023, doi:10.3390/diseases11010029_

Round 1

Reviewer 1 Report

This article summarizes the results of recent clinical studies around the world using vitamin D and calcium in bone metabolic disease. Although not novel, I believe it is an important summary of clinical findings. In addition, I do not see any particular problem with the text, and I think it is acceptable for this journal.

Reviewer 2 Report

Dear Editor,   

thanks for the opportunity to revise the work "Vitamin D and Calcium in osteoporosis, and the role of bone turnover markers: a narrative review of recent data" proposed by Dr. Gavriela et al.   The review aimed to to summarize the effect of vitamin D and calcium supplementation separately and in combination, on bone density, fracture risk and changes of bone turnover markers.   This review is interesting for the scientific community, it is novel and timely. The manuscript is well written and the contents are accurate.

I have only minor comments in order to improve the manuscript: - since the paper summarized the data reported only by RCTs, I suggest to include in the title "..recent data from RCTs" - Page 18 line 519, the authors should complete this sentence mentioning not only the unidirectional negative effect of BMI on vitamin D metabolism, but should mention also the possible intrinsic beneficial metabolic effect of vitamin D itself in overweight and obese patients, extensively reported to influence adiposity and metabolic outcomes [DOI: 10.3390/nu14091816] - Page 18 line 573. Again, the known extraskeletal effects of vitamin D also related to sarcopenia [DOI: 10.3390/nu14091816]  

Thanks.

Reviewer 3 Report

This study summarized the effect of vitamin D and calcium supplementation separately and in combination, on osteoporosis. I have some concerns:

1. Although an abbreviation list has been given at the end of the text, I suggest the author give the full name of the first abbreviation.

2. Table 1 and Table 2 list the authors and years of references, which is inconsistent with the format of the text.

3. A total of 8 studies measured the concentration of bone turnover markers in the 269

plasma or serum of participants [19,36,39,42,43,45,47,49,50]. Here should be 9 studies?

4. It is best to provide a picture overview of the main conclusions of the full text.
